# Modified Ethylsilicates as Efficient Innovative Consolidants for Sedimentary Rock

**Monika Remzova [1,2], Luis A. M. Carrascosa [3], María J. Mosquera [3]**  **and Jiri Rathousky [1,\*]**

[1]  J. Heyrovský Institute of Physical Chemistry of the CAS, Dolejskova 3, 18223 Prague, Czech Republic; monika.remzova@jh-inst.cas.cz

[2]  Department of Physical Chemistry, University of Chemistry and Technology Prague, 16628 Prague, Czech Republic

[3]  TEP-243 Nanomaterials Group, Departamento de Química-Física, Facultad de Ciencias, Universidad de Cádiz, 11510 Puerto Real (Cádiz), Spain; luis.martinez@uca.es (L.A.M.C.); mariajesus.mosquera@uca.es (M.J.M.)

\*   Correspondence: jiri.rathousky@jh-inst.cas.cz; Tel.: +420-26605-3945

**Abstract:** Although silicon alkoxides (especially ethylsilicates) have long been used as consolidants of weathered stone monuments, their physical properties are not ideal. In this study, an innovative procedure for the consolidation of sedimentary rocks was developed that combines the use of organometallic and alkylamine catalysts with the addition of well-defined nanoparticles exhibiting a narrow size distribution centered at ca. 10 nm. As a suitable test material, Pietra di Lecce limestone was selected because of its color and problematic physico-chemical properties, such as rather low hardness. Using the developed procedure, the mechanical and surface properties of the limestone were improved without the unwanted over-consolidation of the surface layers of the stone, and any significant deterioration in the pore size distribution, water vapor permeability, or the stone's appearance. The developed modified ethylsilicates penetrated deeper into the pore structure of the stone than the unmodified ones and increased the hardness of the treated material. The formed xerogels within the stone pores did not crack. Importantly, they did not significantly alter the natural characteristics of the stone.

**Keywords:** ethylsilicates; consolidation; nanoparticles; physico-chemical properties

## 1. Introduction

Historical stone artefacts are exposed to a wide range of outdoor conditions, including freeze-thaw cycles, humidity, irradiation, salt crystallization, and polluted air. As these weathering processes have a negative effect on the mechanical properties of stones, they must be conserved by suitable consolidants.

Alkoxides, especially ethylsilicates and their oligomers [1] are the most frequently used consolidants [2,3] but suffer from several major drawbacks. The syneresis and drying stress during the gelation process cause shrinkage and cracking of the formed xerogel, which reduces its mechanical strength [4]. Due to the presence of pores narrower than two nanometers, so-called micropores, within the consolidant xerogels, the capillary pressure, which is reciprocally proportional to the pore width according to the Young–Laplace equation, is very high [5]. The relatively brittle framework of the siliceous gel cannot resist such a high pressure and cracks. Furthermore, their physical properties, such as the hardness, porosity, and wettability, do not match those of the treated stone.

Because these drawbacks significantly reduce the performance of ethylsilicate consolidants, there has been a great deal of effort to overcome them. For example, gel shrinkage and cracking can be reduced by the formation of wider pores within the gels by suppressing the formation of micropores [6].

Using this approach, the addition of octylamine acting as a catalyst and probably also as a surface-active agent resulted in the formation of gels with an average pore size of approximately 10 nm [7]. Another approach involves improving the structural properties of xerogel by embedding nanoparticles. The modification of gels with nanoparticles has a substantial effect on the physico-chemical and especially mechanical [8] properties of stone. The incorporation of suitable metal oxide nanoparticles was shown to reduce the gel cracking [9]. The embedding of 10–20 nm nanoparticles increased the hardness and Young's modulus of sedimentary stones [10]. Moreover, consolidant xerogels with embedded nanoparticles more closely resemble the properties of natural stones than unmodified gels. Thus, by combining the above-described approaches, it might be possible to develop ethylsilicate consolidants with properties that better meet the requirements for the conservation and preservation of a specific type of stone than the standard ones.

In this study, we developed an innovative procedure for the consolidation of sedimentary rocks that combines the application of organometallic and alkylamine catalysts with the addition of well-defined nanoparticles exhibiting a narrow particle size distribution centered at ca. 10 nm. Using the developed procedure, the mechanical and surface properties of the selected sedimentary rock were improved without the unwanted over-consolidation of the surface layers of the stone, and any significant deterioration in pore size distribution, water vapor permeability, or the stone's appearance was avoided.

The Pietra di Lecce medium-fine grain limestone, selected for evaluating the effectiveness of the developed consolidant, has been a popular building material employed in the construction of historic monuments especially in the city of Lecce, which is therefore commonly nicknamed "The Florence of the South". The most important monuments, dating back to the Baroque era, are the Church of the Holy Cross (Basilica di Santa Croce) and the Lecce Cathedral (Duomo di Lecce).

This stone, already used as a substrate in numerous papers [10–13], shows a yellow-cream color, which enables to determine any changes in color easily. Regarding its composition, it is a Bioclastic limestone with a grain size distribution between 100 and 200 μm. The bioclasts are constituted prevalently by planctonic foraminifera and secondarily by shells. Rare are the quartz and feldspar grains. The binder is abundant and is constituted by microsparitic calcitic matrix. The macroporosity is relevant and due mainly to empty foraminifera chambers. Glauconite neoformation in form of tiny spherulae (80–100 μm) often filling the foraminifera chambers is frequent (5%). Its low hardness and high porosity (34% in total) are optimal for the consolidation and mechanical testing.

## 2. Materials and Methods

### 2.1. Materials

Following chemicals were used: Dynasylan®40 (Evonik, Essen, Germany, hereafter DYN40), hydroxyl-terminated polydimethylsiloxane (ABCR, Karlsruhe, Germany, hereafter PDMS), hydrophilic silica nanoparticles AEROSIL A200 (Evonik, hereafter A200), hydrophobic silica nanoparticles AEROSIL R805 (Evonik, hereafter R805), titania nanoparticles VP AEROPERL P25/20 (Evonik, hereafter VP), *n*-octylamine (Sigma-Aldrich, St. Louis, MO, USA), dioctyltin dilaurate (TIB Chemicals AG, Mannheim, Germany, hereafter DOTL), isopropanol (Sigma-Aldrich).

According to its technical data sheet, DYN40 is a mixture of monomeric and oligomeric ethoxysilanes, with an average chain length of approximately 5 Si–O units. PDMS has a polymerization degree of 12 with a molecular mass between 400 and 700, and an OH percentage ranging from 4 wt.% to 6 wt.%. A200 is a fumed silica with an average particle size of 12 nm and BET surface area of around 200 $m^2 \cdot g^{-1}$. R805 is a similar material, whose surface was treated with octylsilane to give it hydrophobic properties. VP is a micro-granulate with an average size of 20 μm, composed of $TiO_2$ NPs of around 20 nm in size.

## 2.2. Preparation of Consolidants

The sols were synthesized as follows:

- DYN40 was dissolved in isopropanol, in the presence of *n*-octylamine.
- Either PDMS, A200, R805, or VP was added.
- The sols were subjected to agitation in an ultrasonic bath for 30 min.

Alternatively, a sol containing the neutral catalyst DOTL instead of *n*-octylamine was prepared for comparative purposes. As most of the ethylsilicate consolidants in the market present a similar composition, this sample can be employed to compare the performance of the novel products with those available in the market.

All consolidants were prepared following the same procedure (an overview of their composition is given in Table 1).

**Table 1.** Composition of consolidant sols. DYN40 and isopropanol were mixed at 50 vol.%. The concentration of additives and catalysts is related to the volume of DYN40.

| Sample | Additive | Additive Proportion | Catalyst | Catalyst Proportion |
|---|---|---|---|---|
| Dd | – | – | DOTL | 1 vol.% |
| Do | – | – | *n*-octylamine | 0.18 vol.% |
| DoP | PDMS | 5 vol.% | *n*-octylamine | 0.18 vol.% |
| DoA | A200 | 3 wt.% | *n*-octylamine | 0.18 vol.% |
| DoR | R805 | 3 wt.% | *n*-octylamine | 0.18 vol.% |
| DoV | VP | 3 wt.% | *n*-octylamine | 0.18 vol.% |

## 2.3. Characterization of Sols and Xerogels

Immediately after being prepared, the rheological properties of the sols were determined using a Brookfield concentric cylinder viscosimeter (model DV-II+ with UL/Y adapter, Middleboro, MA, USA), the experiments being performed at a constant temperature of 25 °C. The test was performed in dynamic mode at a maximum of 60 rpm. The diameter of the spindle was 25 mm.

The size distribution of nanoparticles in the sols was determined using dynamic light scattering (DLS) carried out on a Malvern Zetasizer Nano ZS instrument (Malvern Instruments, Worcestershire, UK).

A morphological study of the sols was carried out by transmission electron microscopy using a JEOL 2010F TEM/STEM microscope (JEOL, Ltd., Tokyo, Japan), equipped with a JEOL high angle annular dark field (HAADF) detector, enabling the acquisition of STEM images.

In order to study the sol-gel transition, the sols were cast on Petri dishes. Gelation occurred spontaneously, and the gels were left to dry until xerogels were obtained.

Textural properties of the xerogels were determined by the analysis of adsorption isotherms of nitrogen at its boiling point (ca. 77 K) using a Micromeritics 3FLEX apparatus (Norcross, GA, USA). The pore width was described using the IUPAC nomenclature, namely micropores, mesopores, and macropores correspond to the width of less than 2 nm, 2–50 nm, and more than 50 nm, respectively.

Fourier transform infrared spectra (FTIR) in attenuated total reflection (ATR) mode were recorded for the xerogel powders using a Schimadzu FTIR-8400 spectrophotometer (Shimadzu Corp., Kyoto, Japan) in the region of 4000–650 cm$^{-1}$ (resolution of 4 cm$^{-1}$). The samples were in contact with the zinc selenide (ZnSe) ATR crystal and the average of 64 scans was obtained for each.

## 2.4. Application on Stone and Evaluation of the Performance

The sols were sprayed for five seconds onto the 5 cm × 5 cm × 5 cm cubes of the Lecce stone using a pressure of $2 \times 10^5$ Pa. The spraying was repeated five times for each sample. Typically, 2.5 g of the sol was applied on the upper surface of the stone cube. With respect to application on larger area surfaces, the spray technique possesses substantial advantages in comparison with brushing or immersion.

Similar products have been previously applied on pure limestones [14]. The results obtained demonstrate adequate compatibility and adherence to the limestone, producing long-term effectiveness, homogeneity, and continuousness of the coating with a suitable depth of penetration. This compatibility can be explained in terms of the presence of effective interactions between *n*-octylamine integrated in the product and the carbonate substrate.

### 2.4.1. Stone–Product Interaction

The samples were weighed before and immediately after the application of the consolidant to calculate the uptake. In addition, they were re-weighed after complete drying (1 month after the consolidant application) to calculate the dry mass. All the experiments to determine the effectiveness of the products on the stone were carried out 1 month after the consolidant application.

Its penetration into the pore structure of the stone was determined by cutting the stones into thin slabs and dropping water on the cross-section. Dry and wet zones were observed, the latter corresponding to the depth of penetration.

The topography of the untreated and treated samples was studied by scanning electron microscopy, using a JEOL JSM-6510lv microscope.

### 2.4.2. Effectiveness of the Products on the Stone

The enhancement of the mechanical properties due to consolidation was analyzed by several techniques. A Drilling Resistant Measuring System (DRMS, from SINT Technology, Calenzano, Italy) was employed to test the effectiveness of the consolidation, with respect to the untreated sample. Drill bits of 4.8 mm diameter were used with a rotation speed of 200 rpm and penetration rate of 10 mm/min. Nine drillings were performed in three stone samples for each treatment.

By a peeling test [15], the increase in cohesion between the grains of the stone due to the treatment was determined. Furthermore, the adherence of the applied consolidant to the surface of the stone was measured by this method, which is determinant for the treatment durability. The test was carried out by attaching and detaching a Scotch® Magic™ Tape (3M, Maplewood, MN, USA). The process was repeated three times, the mass of material removed was determined by weighing.

Finally, the Vickers hardness test was performed using a Universal Centaur RB-2/200 hardness tester (Innovatest Europe BV Manufacturing, Maastricht, The Netherlands). The loading was 30 kg during 30 s, with a preloaded time of 15 s. For each sample, nine measurements were carried out.

The wettability of the untreated and treated samples was evaluated by measuring static and dynamic (advancing and receding) contact angles of water droplets, according to the procedure published in [16,17]. The water absorption by capillarity (WAC) was measured according to UNE-EN 1925:1999 [18]. After finishing the WAC test, the samples were dried at 60 °C for 24 h. Afterwards, the static and dynamic contact angles were measured again, in order to test the resistance of the treatments to a long-term contact with water.

### 2.4.3. Negative Effects Induced by the Applied Products

Water vapor permeability was determined using an automatic setup developed at the laboratory of the University of Cadiz Nanomaterials Group [19] based on the standard cup test (in accordance with a ASTM E96-90:1990 [20]) in 4 cm × 4 cm × 1 cm slabs. After drying at 60 °C until constant weight, the samples were placed as a cover over the cup, in which moisture saturated at ambient conditions (RH 98%) was maintained. The specimen cup perimeters were sealed with a silicone paste. The climatic chamber was maintained at 23 °C. At the start of tests, low relative humidity was achieved in the chamber by means of a desiccating agent (silica gel). Under these conditions, the humidity gradient across the specimen promoted water vapor flux. The monitoring of the cup mass decrease permitted the progress of vapor transport to be determined, with the temperature and relative humidity in the chamber being registered during the tests. After an initial increase, the relative humidity in the climatic chamber was stabilized at around 35%.

The changes in color were evaluated using a solid reflection spectrophotometer, Colorflex model, from Hunterlab (Reston, VA, USA), calibrated to a white tile standard surface, using Illuminant D65 and observer CIE 10°. Color variations in the CIE *L*a*b** color scale were evaluated using the total color difference (Δ*E**) [21]. Five measurements per sample were carried out.

## 3. Results and Discussion

The former part of this section is devoted to a detailed study into the properties of consolidant sols and xerogels obtained from these sols. The latter part deals with the performance of the sols as consolidants of the Pietra di Lecce limestone with a special attention devoted to the assessment of their suitability for the conservation praxis.

### 3.1. Characterization of the Consolidants

Immediately after being synthesized, the rheological properties of the sols were investigated. Shear stress vs. shear rate curves showed a Newtonian behavior (the regression linear coefficient higher than 0.99). The viscosities of the individual sols calculated from the slope of the curves are shown in Table 2.

**Table 2.** Properties of sols and gels.

| Consolidant | Viscosity (mPa·s) | Gel Time (h) | Appearance | Stability (month) |
|---|---|---|---|---|
| Dd | 2.75 | 60 | Cracked | >18 |
| Do | 2.99 | 48 | Monolithic | >18 |
| DoP | 3.59 | 48 | Monolithic | =12 |
| DoA | 4.64 | 48 | Monolithic | >18 |
| DoR | 4.41 | 48 | Monolithic | >18 |
| DoV | 3.23 | 48 | Phase separation | >18 |

All the sols presented viscosities lower than 5 mPa·s (value for DYN40). This is obviously due to the dissolution in isopropanol. Thus, all the sols are suitable for application on building materials. Previous papers used similar sols without solvents because the viscosity of the sols is enough low (around 5 mPa·s) to assure the product penetration [22]. In this case, we diluted the sol in isopropanol (50%) in order to obtain: (i) better penetration into the porous system in the stone; (ii) suppression of the over consolidation of the stone; (iii) faster hydrolysis due to miscibility of TEOS, water, and isopropanol.

The lowest viscosities were measured for the sols without additives (Dd and Do). They presented similar values, being slightly higher for Do. The addition of PDMS (DoP) increased the viscosity due to the starting co-condensation of PDMS and DYN40, as discussed in previous papers [21,22]. For sols containing NPs, the increase in the viscosity is explained by the presence of a particulate solid. As expected, the sols with the smaller NPs (DoA and DoR) showed the highest viscosities because their high specific surface area (about 200 $m^2 \cdot g^{-1}$) promoted a more extensive aggregation comparing to the product with titania NPs (DoV), whose surface area is around 50 $m^2 \cdot g^{-1}$.

Particle size distributions of the sols without and with added $SiO_2$ and $TiO_2$ nanoparticles measured by DLS are shown in Figure 1. Sols without added nanoparticles (i.e., Dd, Do, DoP) are characterized by the particle sizes in the range 1–10 nm together with some proportion of larger ones 0.1–1 μm in size (maybe dust particles). The nanoparticle-containing sols are characterized by much larger particles 0.06–6 μm in size, which are clearly agglomerates of the primary nanoparticles. The degree of agglomeration seems to depend on the nature of particles added. DoR and DoV exhibit similar particle size distributions in the range from 1 to 2 μm. They do not practically contain any substantially smaller particles. The particle size distribution of DoA containing hydrophilic silica NPs markedly differs, being characterized by a very broad range of particle size from 70 nm to several micrometers. Three distinct maxima were observed for this sample, centered at about 120, 500, and 5000 nm.

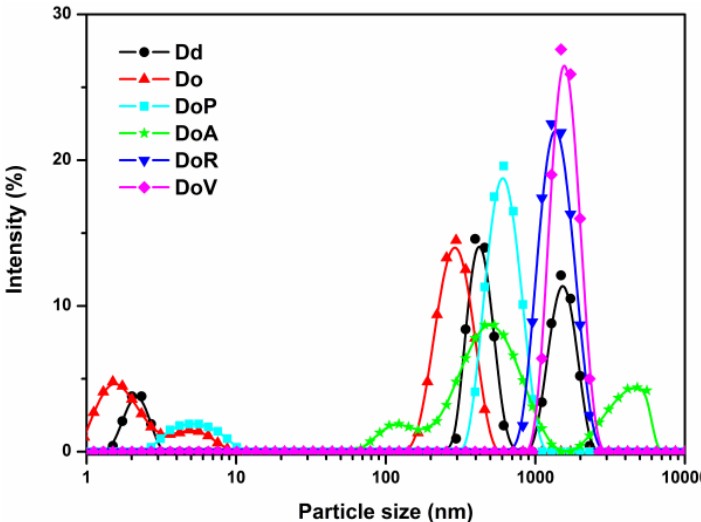

**Figure 1.** The particle size distribution of the sols.

The TEM image of DoR, Figure 2, containing octylated nanoparticles, clearly shows the presence of these nanoparticles which are less than 10 nm in size and exhibit a higher electron density than the sol-gel material around them, which is in accordance with previous works [16,23]. These small nanoparticles are bound into irregularly shaped agglomerates. Generally, the data obtained by DLS and TEM are difficult to compare as the samples change in time and are exposed to different conditions—in sol for DLS and in solid state after the vaporization of the solvent for TEM.

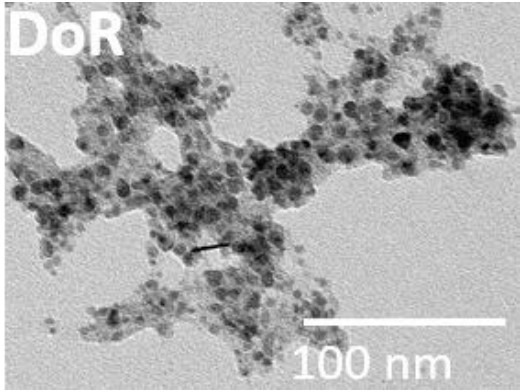

**Figure 2.** TEM image of the DoR sample.

In order to compare the chemical bonds formed, the sols and powders were analyzed by ATR FTIR spectroscopy [24,25] (Figure 3a,b). As the materials are chemically the same, the sols (Figure 3a) show similar spectra. Owing to the presence of alkyl groups either bonded to the siloxane skeleton or in the solvent, a number of bands were observed (designated as B, C, D, E, H). In the range from 3000 to 2800 cm$^{-1}$, three bands (B) correspond to the stretching of carbon–hydrogen (C–H) bonds of alkyl chains [26]. The bands in the range from 1480 to 1385 cm$^{-1}$ (C) are due to the stretching of carbon–hydrogen (C–H) bonds of the alkyl chain, i.e., in the -CH$_3$ and -CH$_2$- groups. Band at 1162 cm$^{-1}$ (E) corresponds to CH$_3$–(C), those at 1255 and 850 cm$^{-1}$ (D1 and D2) to -CH$_3$ bending and rocking in -Si(CH$_3$)$_3$, which is the end group of the PDMS chain. The band at 889 cm$^{-1}$ (H) is due to the deformation of CH$_3$ and CH$_2$ groups of the isopropanol molecule.

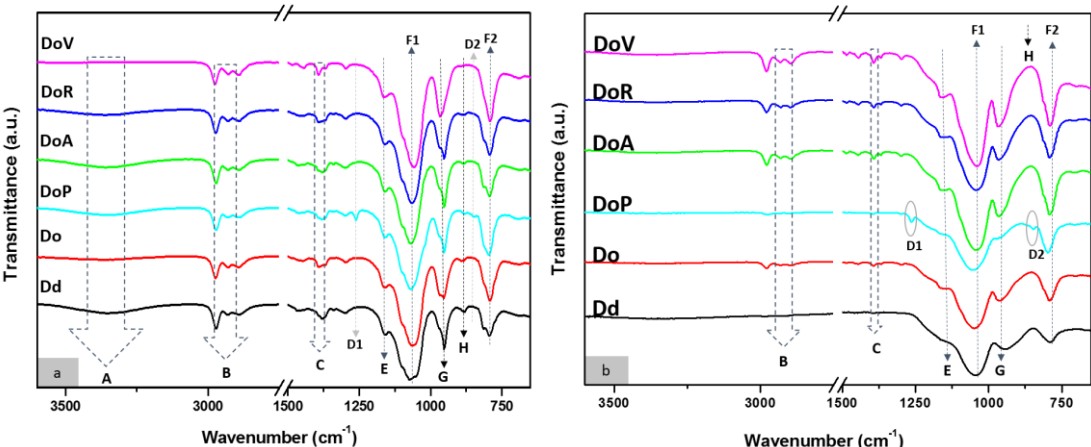

**Figure 3.** FTIR spectra of sols (**a**) and powders (**b**) of ethylsilicate consolidants with and without particles.

Bands at 1045 and 795 cm$^{-1}$ (F) are associated with the growing siloxane skeleton and the creation of a high molecular weight silica polymer [27]. The band (G) is assigned to the Si–OH stretching vibration in the wet gel. FTIR spectrum also shows a peak with low intensity at around 3000–3700 cm$^{-1}$ (A) attributed to the stretching mode of surface silanols (Si–OH), which can provide hydrophilic capability for water absorption due to the formation of hydrogen bridges.

All the products showed high stability over the period of more than 18 months, with the exception of DoP that gelled in a closed vessel after 12 months. Such storage time is sufficient from the point of view of practical application of these consolidants.

The gel time for the sols casted on a Petri dish and exposed to laboratory conditions are shown in Table 2. Generally, a faster gelation was found if octylamine was used as a catalyst, in comparison with DOTL, which is due to the stronger basicity of the amine. No effect of the addition of nanoparticles or PDMS was observed. The longer gel time is convenient from the application point of view as the consolidant products need to penetrate into the pore structure of the stone. Regarding the appearance of the obtained xerogels, that containing DOTL (Dd) as catalyst was completely cracked, whereas those with *n*-octylamine were monolithic. However, in the case of DoV, a phase separation due to the precipitation of the titania NPs was observed.

The FTIR spectra of xerogels after three months showed some changes in comparison with corresponding sols due to the hydrolysis of ethoxy group and the growth of the siloxane chain (Figure 3). Less significant features were the formation of pores within the gel. The formation of a high molecular weight silica polymer is proved by the intensity of bands at 1045–1162 cm$^{-1}$ (E, F1) and 795 cm$^{-1}$ (F2) [27]. The decrease in the isopropanol concentration due to evaporation is apparent from the gradual disappearance of the band at 889 cm$^{-1}$ (H). Generally, the differences between individual samples are less marked, being connected especially with the different rate of hydrolysis depending on the catalyst used and the presence of particles-bands B because the remaining ethoxy groups are much more pronounced for amine-catalyzed samples and those with embedded particles.

Texture parameters of xerogels were obtained by the analysis of nitrogen sorption isotherms (Figure 4). The isotherm on sample Dd, prepared using DODTL as a catalyst, rises sharply at low relative pressures and reaches a plateau at the relative pressure approaching 1. It belongs to the Type I according to the IUPAC classification [28], which corresponds to microporous materials. All the xerogels prepared with *n*-octylamine exhibit the Type IV isotherms, corresponding to mesoporous materials. For these samples, the hysteresis loops associated with capillary condensation are of the Type H2 common with inorganic oxide gels. The pore structures of these materials are complex made up of interconnected networks of pores of different size and shape [28]. The gel containing titania NPs is also mesoporous, its BET surface area and pore volume being much smaller than those of other materials.

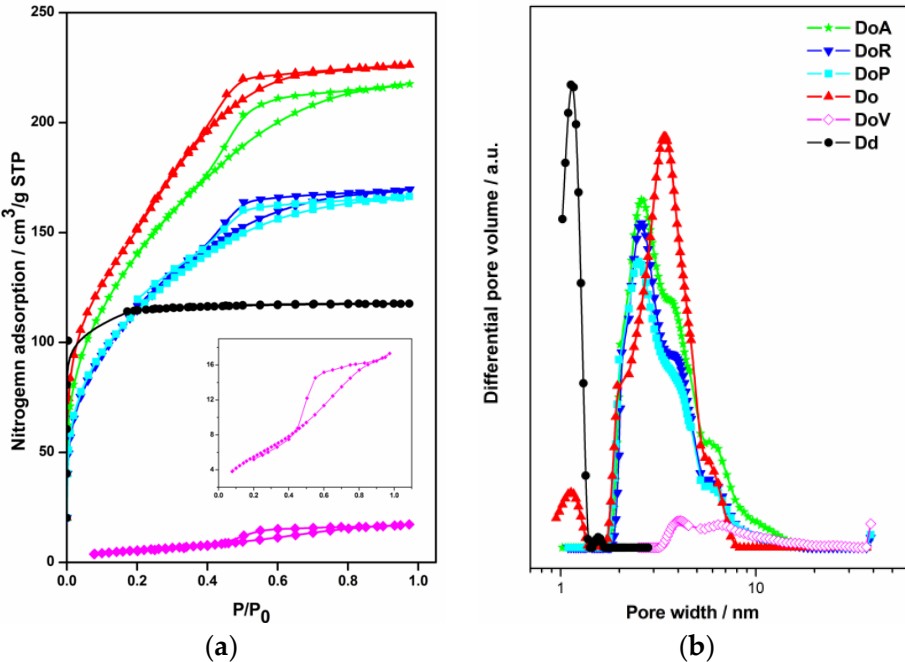

**Figure 4.** Adsorption-desorption isotherms for xerogels (**a**) and the corresponding pore size distributions (**b**) obtained by the NLDFT method.

The pore size distribution of the xerogels was obtained by the Non-Local Density Functional Theory (NLDFT) method (Figure 4b, Table 3). The pore size distribution for the sample Dd is centered at about 1 nm, which is in agreement with its microporous nature. All xerogels synthesized with *n*-octylamine show pores in the mesoporous range. The samples DoP, DoA, and DoR exhibit similar pore size distributions centered at 2.4 nm with a saddle at 3.2 nm. The sample Do exhibits uniform pore size distribution with a maximum at 3.2 nm. Finally, the pores of the DoV sample with a much smaller surface area and pore volume are larger with maxima at 3.5 and about 7 nm.

**Table 3.** Texture parameters of xerogels.

| Consolidant | Micropore Volume (cm$^3 \cdot$g$^{-1}$) | Mesopore Volume (cm$^3 \cdot$g$^{-1}$) | BET Surface Area (m$^2 \cdot$g$^{-1}$) | Pore Width (nm) | Total Porosity (%) ** |
|---|---|---|---|---|---|
| Dd | 0.15 | 0.00 | 1.6 * | 0.8 | 25 |
| Do | 0.00 | 0.35 | 555.0 | 3.2 | 43 |
| DoP | 0.00 | 0.26 | 414.0 | 2.4 | 37 |
| DoA | 0.00 | 0.34 | 514.0 | 2.4 | 45 |
| DoR | 0.00 | 0.26 | 420.0 | 2.4 | 37 |
| DoV | 0.00 | 0.04 | 32.0 | 3.5 | 8 |

\* External surface area; ** calculated from the nitrogen sorption data.

The consistency of the physical properties of xerogels and their structures was tested using correlations developed by Fildalco and Ilharko [29]. Using such a correlation, the percentage of (SiO)$_6$ fold siloxane rings of 10%–12% was assessed for the porosity in the range of 25%–47% (Table 3). From another correlation in [29], it follows that the pore size of xerogels containing 10%–12% of (SiO)$_6$ fold siloxane rings should be in the range of 2.0–2.3 nm, which is in reasonable agreement with the texture data of xerogels given in Table 3. Authors of [29] suggested that hydrophobicity also should depend on the percentages on the siloxane rings. As the percentages of these rings in our samples are similar in a narrow range, a similar range of hydrophobicity should also be expected. The data presented in Table 4 confirm this conclusion as the contact angle for water lies in a narrow range between 120° and 140°.

**Table 4.** A summary of performance of ethylsilicate consolidants.

| Sample | Peeling Test* (mg) | Vickers Hardness Test (kP/mm$^2$) | $\Delta E$ * | Static Angle (°) | Vap. Diffusivity ($\times 10^{-6}$) (m$^2 \cdot$s$^{-1}$) | Porosity (%) | Static Angle after WAC Experiment (°) |
|---|---|---|---|---|---|---|---|
| Untreated | 0.9 ± 0.4 | 15.41 ± 1.76 | – | nd | 3.13 | 37.2 | nd |
| Dd | 0 | 17.5 ± 3.71 | 3.75 ± 1.05 | 0 | 2.71 | 37.1 | nd |
| Do | 0.1 ± 0.1 | 18.91 ± 1.91 | 4.68 ± 1.2 | 140 ± 1 | 2.78 | 40.8 | nd |
| DoP | 0 | 19.43 ± 2.53 | 7.59 ± 0.34 | 149 ± 6 | 2.46 | 42.4 | 135 ± 5 |
| DoA | 0.6 ± 0.1 | 19.10 ± 2.93 | 4.74 ± 0.14 | 144 ± 3 | 3.05 | 47.6 | 125 ± 9 |
| DoR | 0.1 ± 0.1 | 18.44 ± 5.61 | 5.79 ± 0.42 | 129 ± 5 | 3.01 | 40.5 | 113 ± 13 |
| DoV | 0 | 17.06 ± 2.74 | 4.39 ± 0.58 | 120 ± 5 | 3.11 | 43.4 | 108 ± 8 |

* Mass of the removed material; nd, the contact angle could not be measured because of a fast soaking of the water drop.

### 3.2. Application on Stone and Evaluation of the Performance

#### 3.2.1. Stone–Product Interaction

Table 5 shows the uptake, dry-mass, and penetration depth of the products applied on the stone samples. Both uptake and penetration depth depended on the viscosity of the applied sols (Table 2). Do (viscosity 2.99 mPa·s) presented the highest penetration depth (12.5 mm) and uptake (2 wt.%). For the rest of samples, the uptake and penetration are lower because of their higher viscosity. For Dd, the penetration depth could not be measured because of disintegration of the consolidant xerogel in the stone pores (Table 2), which does not allow to identify the two different zones in the cross-section. The evaluation of dry-mass showed values significantly lower than those obtained for uptake due to evaporation of the solvent and hydrolysis of the ethoxy groups. In the case of the samples treated with DoP, this decrease was lower because PDMS does not contain hydrolysable groups.

**Table 5.** Uptake and depth of penetration of the applied consolidants.

| Sample | Uptake (%) | Dry-Matter (%) | Penetration Depth (mm) |
|---|---|---|---|
| Dd | 1.0 ± 0.2 | 0.6 ± 0.1 | – |
| Do | 2.2 ± 0.3 | 1.3 ± 0.2 | 12.5 ± 1.3 |
| DoP | 1.2 ± 0.1 | 0.8 ± 0.1 | 9.1 ± 0.9 |
| DoA | 1.1 ± 0.1 | 0.8 ± 0.1 | 5.4 ± 1.3 |
| DoR | 1.2 ± 0.3 | 0.8 ± 0.2 | 4.1 ± 0.7 |
| DoV | 1.4 ± 0.1 | 0.9 ± 0.1 | 8.4 ± 1.5 |

To investigate the changes in the topography of the stones caused by the consolidation, SEM images were acquired (Figure 5). The untreated sample is mainly composed of grains (Figure 5a). While the surface coated with Dd (Figure 5b) showed cracks, the treatment with Do lead to a crack-free smooth layer covering the surface (Figure 5c). This is due to the formation of a mesoporous material (see the physisorption data, Figure 4 and Table 3), as explained in our previous works [10,19]. Owing to the additives, the roughness of the surface increased. For DoP (Figure 5), this roughness is due to the shrinkage of the xerogel during the drying, because of its high flexibility [24]. For nanoparticle-containing coatings, the roughness is clearly due to the presence of NPs in the composition. The appearances of DoA- and DoR-coated materials (Figure 5e,f, respectively) were similar, since the A200 and R805 nanoparticles had the same size. In the case of DoV, greater aggregates were observed due to the larger VP nanoparticles.

#### 3.2.2. Effectiveness of the Products on Stone

From Figure 6 it is apparent that the untreated sample showed a very low resistance to drilling, which was more or less increased by all the treatments. According to [30], the penetration depth of consolidants should be above 5 mm. In our case, all the products show penetration over 5 mm. For Dd the increase within the first five millimeters was roughly 100%, deeper however practically negligible. This is due to the cracking of the xerogel for sample Dd inside the stone, as seen in the SEM images (Figure 5b).

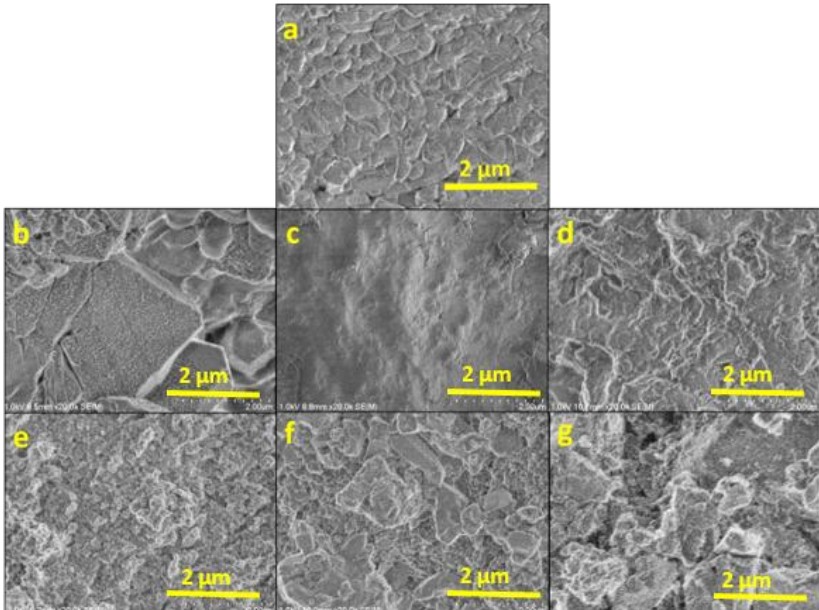

**Figure 5.** SEM images of the surfaces of the untreated stones and their treated counterparts: (**a**) Untreated, (**b**) Dd, (**c**) Do, (**d**) DoP, (**e**) DoA, (**f**) DoR, (**g**) DoV.

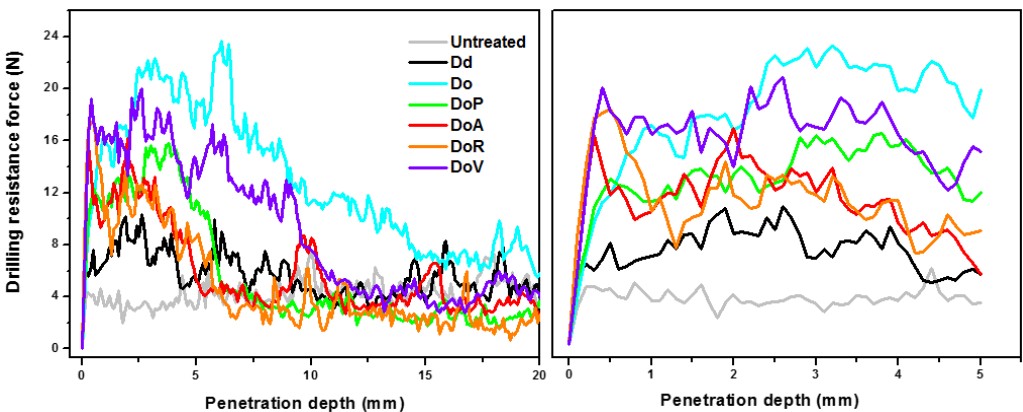

**Figure 6.** DRMS results for the untreated and treated samples.

On the contrary, all the amine catalyzed consolidant sols lead to a more substantial increase in the resistance force and the results are in good agreement with the observed penetration depth (Table 5) and the uptake of the consolidants. For DoP, there was marked increase (up to 200%) within the first 5 mm, in greater depth practically negligible. This is due to the lower penetration of this consolidant because of its higher viscosity. Moreover, the incorporation of PDMS in the xerogel network promotes the change from quaternary siloxane bonds (from DYN40) to a mixture of quaternary and binary (from PDMS) siloxane bonds [15,30], which provide more flexibility to the silica network, allowing the xerogel to shrink.

The highest degree of consolidation among the samples tested was achieved for Do and the increase in resistance persevered even to the depth of 20 mm. Regarding the particle-modified consolidants, the most promising performance was found for DoV doped with titania particles. For this sample, the range of substantial increase in resistance was wider, up to the depth of 10 mm. This sample was only slightly weaker than the leading Do.

The application of DoA- and DoR-doped consolidants caused the lowest increase in the mechanical properties, among the products catalyzed with *n*-octylamine. This is because of two reasons: the lower penetration depth due to their higher viscosity compared with the rest of products, and the very small

size of both NPs (around 12 nm), which are not able to form big aggregates with DYN40, oppositely to the titania NPs (see SEM, Figure 5).

Alternatively to the DRMS experiments, the determination of the Vickers hardness was performed (Table 4). The obtained data are in a reasonable agreement with the DRMS experiments. While the untreated sample of Lecce stone exhibited a hardness of $15.41 \pm 1.76 \text{ kP/mm}^2$, the treatment with octyl amine catalyzed consolidant let to an increase to $18.91 \pm 1.91 \text{ kP/mm}^2$. The stones treated with nanoparticle modified consolidants exhibited even higher hardness, such as $19.10 \pm 2.93 \text{ kP/mm}^2$ for DoA.

Table 4 shows also the results obtained from the peeling tests. The results demonstrate that all the treatments decrease the amount of material removed by the adhesive tape, confirming the consolidation ability. With the exception of DoA, with all the other consolidants the mass of removed stone fragments was practically negligible. With the sample DoA the cohesion of the surface layers was only slightly better than that of the untreated stone.

Concerning the wettability of the treated samples, Table 4 shows that the contact angle for water substantially increased due to the consolidation, while the untreated stone exhibited a very fast soaking of water.

Actually, such high hydrophobicity was to be expected only for samples containing a hydrophobic component (i.e., DoP and DoR). Clearly, the non-hydrolyzed ethoxy groups from DYN40 fundamentally influenced the wetting properties of the consolidated stones. As these groups decrease the surface energy, they have a hydrophobic effect and function as the hydrophobic components themselves. The hydrophobicity of the Lecce stone is clearly enhanced due to its surface roughness according to the Wenzel model. For DoV, the contact angle is slightly lower than for other particle-modified samples, probably due to the hydrophilic nature of titania particles.

In previous papers from our group [16,17], the addition of the $SiO_2$ nanoparticles to a hydrophobic material increased its hydrophobic performance and promoted water repellence properties due to the creation of a novel roughness on the treated surface. However, in the present study, this effect is negligible as the product is applied to stones with high roughness [25].

The water absorption by capillarity (WAC) is presented in Figure 7. For the untreated stone, the inset shows a very fast increase in the mass of the absorbed water within the first hour, followed by a practically horizontal plateau. For the comparative sample Dd, the mass of absorbed water is only slightly lower because of substantial disintegration of the consolidant xerogels in the stone pores (Figure 5b). However, for consolidants catalyzed by octylamine, the water absorption is drastically reduced, practically by two orders of magnitude. The products with PDMS or R805 (DoP or DoR) are the most effective in decreasing water absorption, confirming their hydrophobic properties. For the stone treated with the consolidant Dd, the decrease in water absorption by capillarity was only about 7%, but for all the samples treated with the novel consolidants it reached 98%–99%.

Finally, the contact angles were measured again after the WAC test, in order to determine the degree of the hydrolysis of surface ethoxy-groups due to a long contact time with water (Table 4). A complete loss of the hydrophobicity was found for both Dd and Do samples, i.e., for those without any dopant. These two samples exhibited a rapid absorption of the water droplets. This is due to the hydrolysis of the ethoxy groups to hydroxyls during the contact with water. DoP and DoR do not show significant variation in contact angles, demonstrating that the hydrophobic behavior of these products is due to the action of PDMS [31,32] or R805 NPs, respectively, reducing the surface energy.

As DoA and DoV do not have any low surface energy component added, their hydrophobic behavior is due to the presence of non-hydrolyzed ethoxy groups. Actually, after long-contact with water, these groups should be hydrolyzed as for Do. This contradictory observation can be explained according to Wang et al. [33]. These authors prepared superhydrophobic coatings on glass controlling the ratio of hydrolysable (ethoxy) and pre-hydrolyzed (hydroxyl) groups. The surfaces were placed under a water rain column for 2 h, and the static contact angle was measured before and after the test. Results showed a lower decrease in the contact angle for the surfaces with higher proportion

of hydroxyl groups. XPS spectra demonstrated that water preferably attacks OH groups (due to its affinity) rather than ethoxy groups. Thus, hydroxyl groups protect ethoxy groups from being hydrolyzed. In our case, both DoA and DoV have hydroxyl groups (from silica or titania NPs, respectively), which interact with water, preventing the hydrolysis of the remaining ethoxy groups from DYN40.

There is some change in color due to the consolidation (Table 4). Depending on the catalysts used, a slightly greater change was observed for octylamine in the comparison with DOTDL, both being within the limits of the generally acceptable color changes. Also, for particle-modified consolidants, the change was within the limits of acceptability. The most significant change was observed for DoP, i.e., for the consolidant with added PDMS. As the change in color was outside the acceptance limits, there is a need for caution when applying this type of consolidant.

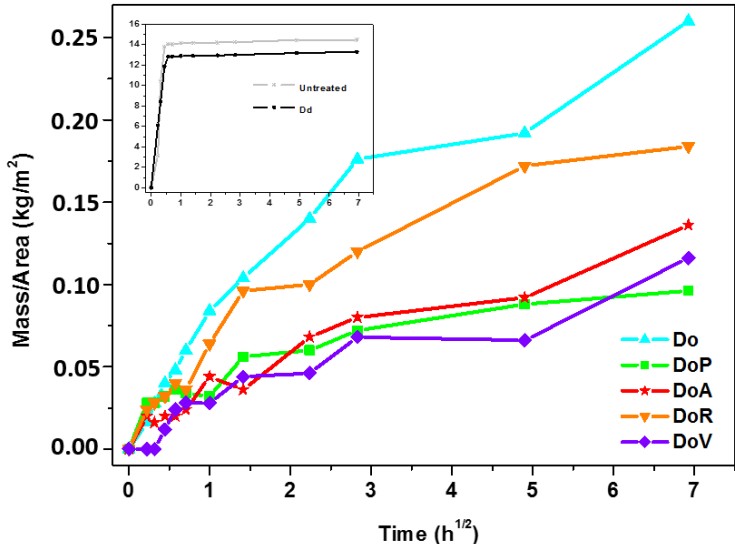

**Figure 7.** Water absorption by capillarity for stone samples treated with consolidants.

## 4. Conclusions

To sum up, the systematic study has shown that the effect of the catalyst selected and the nanoparticles or PDMS used as additives on the properties of ethylsilicate consolidants is rather complex, which enables a fine tuning of their performance. Especially the character of the formed porosity with the closely related xerogel cracking depends on the catalyst used and the presence and surface properties of nanoparticles, which also influence the rate of gelation. It is a major advantage of using octylamine as catalysts that the treated stone exhibits a smooth surface without cracks.

Another important functional property, namely the depth of the consolidation, is beneficially influenced by the use of octylamine as a catalyst instead of organometallic compounds, which are most commonly used. Interestingly, the performance concerning the depth of penetration of the consolidant with $TiO_2$ nanoparticles added is very similar to the undoped one. As these particles additionally suppress the cracking and provide a consolidant with photocatalytic functionality, this type of consolidant is definitely a promising choice for a range of applications.

The wettability and water absorption by capillarity of the treated stones depend on catalysts used as well as on the character of nanoparticles used as a dopant. As this property may be desirable, the method developed in this study provides an efficient way to control it.

Finally, because our modified ethylsilicate consolidants are based on inorganic compounds, they do not significantly alter the natural characteristics of the treated stone, which is very promising for the application in the conservation praxis. To determine the applicability of the developed consolidants, a study into their performance on various types of weathered stones is in progress, including testing the durability of the consolidation treatment by artificial accelerated weathering.

**Author Contributions:** Conceptualization, J.R. and M.J.M.; Methodology, J.R. and M.J.M.; Validation, M.R. and L.A.M.C.; Investigation, M.R. and L.A.M.C.; Writing-Original Draft Preparation, M.R.; Writing-Review & Editing, J.R.; Visualization, M.R.; Supervision, J.R. and M.J.M.; Project Administration, J.R.; Funding Acquisition, J.R.

**Funding:** This research was funded by the Czech Science Foundation (No. 17-18972S). The authors are also grateful to the Spanish Government/FEDER-EU (MAT2013-42934-R, MAT2017-84228-R and IPT-2012-0959-310000).

**Acknowledgments:** The authors acknowledge the assistance provided by the Research Infrastructures NanoEnviCz (Project No. LM2015073) supported by the Ministry of Education, Youth and Sports of the Czech Republic and the project Pro-NanoEnviCz (Reg. No. CZ.02.1.01/0.0/0.0/16_013/0001821) supported by the Ministry of Education, Youth and Sports of the Czech Republic and the European Union-European Structural and Investments Funds in the frame of Operational Programme Research Development and Education. Luis A.M. Carrascosa thanks the Spanish Government for a grant associated to the project IPT-2012-0959-310000.

**Conflicts of Interest:** The authors declare no conflict of interest.

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
