# Peer review of "Modified Ethylsilicates as Efficient Innovative Consolidants for Sedimentary Rock"

_coatings, doi:10.3390/coatings9010006_

Reviewer 1 Report

This manuscript, regarding innovative nano-filled ethylsilicates for consolidation of a limestone, i.e. Pietra di Lecce, is very interesting in terms of scientific contents and novelty. The paper is well written, the research has been conducted with a proper approach and the results are presented in a clear and critical way. Moreover, the work matches into the scope of the journal.

For the mentioned reasons, in my opinion it can be accepted for publication. It would benefit from some minor improvements,  following listed.

Since the non-European readers could be not familiar with the selected stone substrate, i.e. Pietra di Lecce, the Authors should supply additional information on its use in constructions and Cultural Heritage artworks, where it can be found and extracted and some peculiar properties.

Referring to the rheological tests, it would be advisable to add if these were performed in steady or dynamic mode (I suppose the latter), with indication of test parameters (lines 93-95).

In the “Results and discussion” section, it is suggested to add some comments on the additional features brought about by the presence of a nano-filler in a protective hydrophobic coating, even referring to previous pertinent  literature.

For the proceeding of the research it is, finally, suggested to analyze the mechanical behavior of the consolidated stone specimens, with and without consolidants, also with different standardized tests, such as mechanical tests performed in compression mode.

Author Response

The answers to reviewer’s comments

The authors are grateful to all three reviewers for valuable comments and suggestion which have definitely improve the quality of the manuscript, especially with respect to the precision and completeness of formulations and readability for the wider range of the readership.

We have carefully addressed all the suggestions and modified the manuscript if needed. Our detailed answers are given below.

REVIEWER No. 1

1.1 Since the non-European readers could be not familiar with the selected stone substrate, i.e. Pietra di Lecce, the Authors should supply additional information on its use in constructions and Cultural Heritage artworks, where it can be found and extracted and some peculiar properties.

The following paragraph was added to the Introduction:

The Pietra di Lecce medium-fine grain limestone, selected for evaluating the effectiveness of the developed consolidant, has been a popular building material employed in the construction of historic monuments especially in the city of Lecce, which is therefore commonly nicknamed "The Florence of the South". The most important monuments, dating back to the Baroque era, are the Church of the Holy Cross (Basilica di Santa Croce) and the Lecce Cathedral (Duomo di Lecce).

This stone, already used as a substrate in numerous papers [10,11,2], shows a yellow-cream colour, which enables to determine any changes in colour easily. Regarding its composition, it is a Bioclastic limestone with a grain size distribution between 100 and 200 µm. The bioclasts are constituted prevalently by planctonic foraminifera and secondarily by shells. Rare are the quartz and feldspar grains. The binder is abundant and is constituted by microsparitic calcitic matrix. The macroporosity is relevant and due mainly to empty foraminifera chambers. Glauconite neoformation in form of tiny spherulae (80-100 µm) often filling the foraminifera chambers is frequent (5%). Its low hardness and high porosity (34% in total) are optimal for the consolidation and mechanical testing.

1.2 Referring to the rheological tests, it would be advisable to add if these were performed in steady or dynamic mode (I suppose the latter), with indication of test parameters (lines 93-95).

We have added following paragraph to the Experimental section:

The test was performed in dynamic mode at a maximum of 60 rpm. The diameter of the spindle was 25 mm.

1.3 In the “Results and discussion” section, it is suggested to add some comments on the additional features brought about by the presence of a nano-filler in a protective hydrophobic coating, even referring to previous pertinent  literature.

We have added following paragraph to the Results and Discussion section:

In previous papers from our group [16,17], the addition of the SiO2 nanoparticles to a hydrophobic material increased its hydrophobic performance and promoted water repellence properties due to the creation of a novel roughness on the treated surface. However, in the present study this effect is negligible as the product is applied to stones with high roughness [23].

1.4 For the proceeding of the research it is, finally, suggested to analyze the mechanical behaviour of the consolidated stone specimens, with and without consolidants, also with different standardized tests, such as mechanical tests performed in compression mode.

We have added following paragraph to the Results and Discussion section:

Alternatively, to the DRMS experiments, the determination of the Vickers hardness was performed (Table 5). The obtained data are in a reasonable agreement with the DRMS experiments. While the untreated sample of Lecce stone exhibited a hardness of 15.41±1.76 kP/mm2, the treatment with octyl amine catalyzed consolidant let to an increase to 18.91±1.91 kP/mm2. The stones treated with nanoparticle modified consolidants exhibited even higher hardness, such as 19.10±2.93 kP/mm2 for DoA.

Reviewer 2 Report

Comments to the Author:
The authors of this paper present an interesting procedure for the consolidation of sedimentary rocks, which combines the use of organometallic and alkylamine catalysts with the addition of well-defined nanoparticles. Nevertheless, some details should be considered by the authors:

Introduction

COMMENT: Page 1, lines 34-42: more recent references should be added.

COMMENT: Page 2, lines 53-54: references should be added.

COMMENT: Page 2, lines 60-62: long sentence. Please rephrase.

COMMENT: Page 2, line 66: How this porosity value was estimated? Please comment.

Materials and Methods

COMMENT: Page 2, line 88: Substitute “given” for “is given”.

COMMENT: Page 3, line 107-109: more experimental details need to be added about the ATR crystal type, number of scans and how the spectra were processed.

Results and discussion

COMMENT: Page 4, line 162: Substitute “into” for “of”.

COMMENT: Page 5, line 184: Substitute “of sols” for “of the sols”.

COMMENT: Page 6, line 205 (figure caption): TEM images of Do or DoR samples? Please clarify. It is suggested to add more TEM Images.

COMMENT: Page 6, lines 206-220: References have to be added for the FTIR analysis.

COMMENT: Page 8, line 271: Substitute “…of this rings in ours samples is…” for “…of these rings in ours samples are…”.

COMMENT: Page 9, Table 4: The number of decimal digits for the value and the error must be the same (for example, substitute 1.19±0.3 for 1.2±0.3).

COMMENT: Page 11, Table 5: The number of decimal digits for the value and the error must be the same.

Author Response

The answers to reviewer’s comments

The authors are grateful to all three reviewers for valuable comments and suggestion which have definitely improve the quality of the manuscript, especially with respect to the precision and completeness of formulations and readability for the wider range of the readership.

We have carefully addressed all the suggestions and modified the manuscript if needed. Our detailed answers are given below.

REVIEWER No. 2

Introduction

2.1 COMMENT: Page 1, lines 34-42: more recent references should be added.

We have added more references to the manuscript.

2.2 COMMENT: Page 2, lines 53-54: references should be added.

The sentence in the manuscript formulates the actual aim of our research which has not been studied earlier. There are only references concerning either the creation of wider pores or the change in the mechanical properties of the consolidant due to incorporation nanoparticles. Therefore no references could be added to the manuscript.

2.3 COMMENT: Page 2, lines 60-62: long sentence. Please rephrase.

We have rephrased the sentence:

Using the developed procedure, the mechanical and surface properties of the selected sedimentary rock were improved. In addition, the any unwanted over-consolidation of the surface layers of the stone, and any significant deterioration in pore size distribution, water vapour permeability or the stone’s appearance were avoided.

2.4 COMMENT: Page 2, line 66: How this porosity value was estimated? Please comment.

The porosity was determined from the pore volume determined by the analysis of adsorption isotherms of nitrogen at the boiling point of liquid nitrogen (ca. 77 K) and the volume of solid material, according to the formula porosity in per cent = pore volume / (pore volume + volume of the solid material) x 100.

2.5 COMMENT: Page 2, line 88: Substitute “given” for “is given”.

We have changed the sentence.

2.6 COMMENT: Page 3, line 107-109: more experimental details need to be added about the ATR crystal type, number of scans and how the spectra were processed.

We have supplemented the require data to the text.

2.7 COMMENT: Page 4, line 162: Substitute “into” for “of”.

We have modified the manuscript according to the suggestion.

2.8 COMMENT: Page 5, line 184: Substitute “of sols” for “of the sols”.

We have modified the manuscript according to the suggestion.

2.9 COMMENT: Page 6, line 205 (figure caption): TEM images of Do or DoR samples? Please clarify. It is suggested to add more TEM Images.

It is TEM image of the DoR sample. As the images are very similar and do not bring much new information, we have decided not to add another image.

2.10 COMMENT: Page 6, lines 206-220: References have to be added for the FTIR analysis.

We have added references for the FTIR analysis.

2.11 COMMENT: Page 8, line 271: Substitute “…of this rings in ours samples is…” for “…of these rings in ours samples are…”.

We have modified the manuscript according to the suggestion.

2.12 COMMENT: Page 9, Table 4: The number of decimal digits for the value and the error must be the same (for example, substitute 1.19±0.3 for 1.2±0.3).

We have modified the table according to the suggestion.

2.13 COMMENT: Page 11, Table 5: The number of decimal digits for the value and the error must be the same.

We have modified the table according to the suggestion.

Reviewer 3 Report

The article is an study of the consolidation effect of developed consolidation products that combine the use of organometallic and alkylamine catalysts with the addition of nanoparticles in a sedimentary rock. The consolidation effect is studied analyzing the mechanical and surface properties of the treated limestones. Although the article is well written and present novel and interesting results some things have to be improve in my point of view:

Materials and methods section:

- The amount of applied consolidation products is not indicated in the “Application on stone and evaluation of the performance” section. The quantity of applied product is very important in order to study the consolidation effect of each product.

- What is the concentration of the product? 50%? Why? Could it be too concentrated to achieve the penetration of the product into the porous system?

- The selection of the spray method to apply the consolidation products is not argued. Why was it selected between other more used application methods like for example brushing?

- How may stone samples were used for study the consolidation effect of each consolidation product? The number of stone samples that have been used in the study is not indicated. This is a key data to be able to analyze the representativeness of the results.

- It will be useful to add to the study a commercial ethyl silicate consolidation product in order to be able to compare the results of this novel developed products with those obtained by a commercial currently used one

- How many drilling were made in each stone?

- In my point of view, to study only the static and dynamic contact angles is not enough to analyze the resistance of the treatments to a long-term contact with water and to assess the durability of the applied consolidation products.

Results and discussion section:

            - Line 243: “3” is missing, change the figure caption by “Figure 3”

- In the results a very low penetration deep of the applied consolidation product is obtained so, only surface consolidation is achieved ant not in deep consolidation. This should be highlighted in the results and conclusions.

- Line 302: Figure 5. Describe in the caption what each image is (5a……/ 5b……)

- Figure 6. Differences on the graphic are not easy to observer. Maybe it could be clarifying to change the line type of each measure

- Line 309: specify what images is from Figure 5

- Table 5: It would be clarifying to calculate the produced changes in percentage to be able to compare the date. For example, the decrease in the realized material of the peeling test or the increase in the Vickers hardness.

- Although the water absorption by capillarity of the samples are show in Figure 7, the date of the absorbed water quantity and missing. The date of the reduction in the absorbed water quantity are necessary. Add the date into the Table 5 and comment the results in the argumentation.

- The porosity values are higher in the treated samples. Why is that? There is no argumentation about that date in the text. How many analysis where done in each consolidated samples? There have to be several analyses in order to obtain representative data and be able to draw conclusions

- The authors have thought about trying these products in calcite-based substrates?

Author Response

The answers to reviewer’s comments

The authors are grateful to all three reviewers for valuable comments and suggestion which have definitely improve the quality of the manuscript, especially with respect to the precision and completeness of formulations and readability for the wider range of the readership.

We have carefully addressed all the suggestions and modified the manuscript if needed. Our detailed answers are given below.

REVIEWER No. 3

Materials and methods section:

2.1 The amount of applied consolidation products is not indicated in the “Application on stone and evaluation of the performance” section. The quantity of applied product is very important in order to study the consolidation effect of each product.

We have added following sentence to the Application on stone and evaluation of the performance section:

Typically, 2.5 g of the sol was applied on the upper surface of the stone cube 5 cm × 5 cm in size.

2.2 What is the concentration of the product? 50%? Why? Could it be too concentrated to achieve the penetration of the product into the porous system?

We have added the following paragraph to the Discussion section:

All the sols presented viscosities lower than 5 mPa·s (value for DYN40). This is obviously due to the dissolution in isopropanol. Thus, all the sols are suitable for application on building materials. Previous papers used similar sols without solvents because the viscosity of the sols is enough low (around 5 mPa·s) to assure the product penetration [20]. In this case, we diluted the sol in isopropanol (50%) in order to obtain: (i)Better penetration in to the porous system in the stone; (ii) The suppression the over consolidation of the stone; (iii) A faster hydrolysis due to miscibility of TEOS, water and isopropanol.

2.3 The selection of the spray method to apply the consolidation products is not argued. Why was it selected between other more used application methods like for example brushing?

We have added the following paragraph to the paper:

With respect to the application on larger area surfaces the spray technique is more advantages to have advantages in comparison with brushing or immersion.

2.4 How may stone samples were used for study the consolidation effect of each consolidation product? The number of stone samples that have been used in the study is not indicated. This is a key data to be able to analyze the representativeness of the results.

We have added following sentence to the Application on stone and evaluation of the performance section:

Penetration rate of 10 mm/min. 9 drilling were performed in three stone samples for each treatment under study.

2.5 It will be useful to add to the study a commercial ethyl silicate consolidation product in order to be able to compare the results of this novel developed products with those obtained by a commercial currently used one

The following sentence was added to the manuscript in order to explain the use of a formulation similar to the commercial products:

Alternatively, a sol containing the neutral catalyst DOTL instead of n-octylamine was prepared for comparative purposes. As most of the ethylsilicate consolidants in the market present a similar composition, this sample can be employed to compare the performance of the novel products with those available in the market.

2.6 How many drilling was made in each stone?

We have added following sentence to the Application on stone and evaluation of the performance section:

Nine drilling were performed in each stone.

2.7 In my point of view, to study only the static and dynamic contact angles is not enough to analyze the resistance of the treatments to a long-term contact with water and to assess the durability of the applied consolidation products.

We have not spoken about the durability of the consolidants. We exclusively discussed that the durability of hydrophobic performance as described in page 377, line 358-364.

2.8 Line 243: “3” is missing, change the figure caption by “Figure 3”

We have changed the caption of Figure 3 according to the remark

2.9 In the results a very low penetration deep of the applied consolidation product is obtained so, only surface consolidation is achieved ant not in deep consolidation. This should be highlighted in the results and conclusions.

We have added following explanation to the manuscript:

From the Figure 6 it is apparent that the untreated sample showed a very low resistance to drilling, which was more or less increased by all the treatments. According to [28] the penetration depth of consolidants should be above 5 mm. In our case, all the products show penetration over 5 mm. For Dd the increase within the first five millimeters was roughly 100%, deeper however practically negligible. This is due to the cracking of the xerogel for sample Dd inside the stone, as seen in the SEM images (Figure 5b).

2.10 Line 302: Figure 5. Describe in the caption what each image is (5a……/ 5b……)

We have added, the following sample designation to the figure caption:

(a) Untreated, (b) Dd, (c) Do, (d) DoP, (e) DoA, (f) DoR, (g) DoV.

2.11 Figure 6. Differences on the graphic are not easy to observer. Maybe it could be clarify in to change the line type of each measure

We have tested several possibilities of the style of the lines. As the main purpose of the figure is to provide a graphic comparison of the performance of individual consolidants, it is not suitable to divide the figure into two ones. The change in the style of curves (using solid, dash or dotted ones, etc) or using different symbols did not proved itself as such modified figure were rather crowded and confused. Therefore we have decided to preserve the original shape of the figure.

2.12 Line 309: specify what images is from Figure 5

We have changed the sentence on lines 308-309 to:

This is due to the cracking of the xerogel for sample Dd inside the stone, as seen in the SEM images (Figure 5b).

2.13 Table 5: It would be clarifying to calculate the produced changes in percentage to be able to compare the date. For example, the decrease in the realized material of the peeling test or the increase in the Vickers hardness.

In order not to have too many columns in Table 5, we have added the following sentence to the paragraph in lines 336-337:

The increase the Vickers hardness for the consolidant with the organometalic catalyst (Dd) was about 13% whereas for those containing octylamine was increased (about 20%). For the two samples DoA and DoR containing based SiO2 NPs, the increase in the Vickers hardness is very similar (24% and 20%, respectively)

2.16 Although the water absorption by capillarity of the samples are show in Figure 7, the date of the absorbed water quantity and missing. The date of the reduction in the absorbed water quantity are necessary. Add the date into the Table 5 and comment the results in the argumentation.

Because the reduction the absorbed water quantity for the novel samples is very similar and substantially different than that for Dd sample, we have added the following sentence:

If quantified, while for the stone treated with the consolidant Dd the decrease in water absorption by capillarity was only about 7%, for all the samples treated with the novel consolidants it reached 98-99%.

2.17 The porosity values are higher in the treated samples. Why is that? There is no argumentation about that date in the text. How many analysis where done in each consolidated samples? There have to be several analyses in order to obtain representative data and be able to draw conclusions

We have added following comment under the table 5:

**calculated from nitrogen sorption data.

2.18 The authors have thought about trying these products in calcite-based substrates?

The following paragraph was included in the manuscript:

Similar products have been previously applied on pure limestones [14]. The results obtained permit to conclude that an adequate compatibility and adherence to the limestone was observed, producing a long-term effectiveness, homogeneity and continuousness of the coating with a suitable depth of penetration. This compatibility can be explained in terms of the presence of effective interactions between n-octylamine, integrated in the product, and the carbonate substrate.

Round  2

Reviewer 1 Report

The Authors took into account the criticisms/suggestions of this reviewer. The paper is now suitable for publication.